# Structural Elucidation of an Atropisomeric Entcassiflavan-(4β→8)-Epicatechin Isolated from *Dalbergia monetaria* L.f. Based on NMR and ECD Calculations in Comparison to Experimental Data

**DOI:** 10.3390/molecules27082512

**Published:** 2022-04-13

**Authors:** Patrícia Homobono Brito de Moura, Wolfgang Brandt, Andrea Porzel, Roberto Carlos Campos Martins, Ivana Correa Ramos Leal, Ludger A. Wessjohann

**Affiliations:** 1Natural Products Research Institute (IPPN), Center of Health Sciences, Federal University of Rio de Janeiro (UFRJ), Rio de Janeiro 21941-902, RJ, Brazil; patricia.homobono@gmail.com (P.H.B.d.M.); roberto.rcc@gmail.com (R.C.C.M.); 2Natural Products and Food Department, Pharmacy Faculty, Center of Health Sciences, Federal University of Rio de Janeiro (UFRJ), Rio de Janeiro 21941-902, RJ, Brazil; 3Department of Bioorganic Chemistry, Leibniz Institute of Plant Biochemistry (IPB), Weinberg 3, 06114 Halle, Germany; wolfgang.brandt@ipb-halle.de (W.B.); aporzel@ipb-halle.de (A.P.)

**Keywords:** *Dalbergia monetaria*, proanthocyanidin, procassidin dimer, electronic circular dichroism (ECD), quantum chemical calculation, NMR

## Abstract

A rare dihydoxyflavan-epicatechin proanthocyanidin, entcassiflavan-(4β→8)-epicatechin, was isolated from *Dalbergia monetaria*, a plant widely used by traditional people from the Amazon to treat urinary tract infections. The constitution and relative configuration of the compound were elucidated by HR-MS and detailed 1D- and 2D-NMR measurements. By comparing the experimental electronic circular dichroism (ECD) spectrum with the calculated ECD spectra of all 16 possible isomers, the absolute configuration, the interflavan linkage, and the atropisomers could be determined.

## 1. Introduction

*Dalbergia monetaria* L.f., popularly known as “veronica”, is a medicinal plant widely used in the Amazon region by traditional people to treat some infectious diseases such as urinary tract infections (UTIs) [1]. Regarding the chemical composition of *D. monetaria*, phenolic compounds, such as isoflavonoids and proanthocyanidins (PAs), were isolated from the bark [2,3]. PAs possess a range of biological effects; among them, antibacterial activity is widely reported, especially against Gram-negative and -positive strains [4,5]. However, condensed tannins have not been widely reported from the *Dalbergia* genus so far. Proanthocyanidin A-2 was identified in the leaves and heartwood crude extracts of *D. boehmii* [6]. Additionally, (2*R*,3*R*,4*R*)-3,3′,4′,7-tetrahydroxyflavan-(4β→8)-epicatechin and (2*R*,3*R*,4*R*)-3,4′,7-trihydroxyflavan-(4β→8)-epicatechin, built from flavan-3-ol units, were isolated from the stembark of *D. monetaria* [3]. Flavan-3-ols are the best-known building blocks of proanthocyanidins, mainly catechin and epicatechin. In contrast to flavanols, flavans are very rare as monomeric units of PAs [7]. Thus, dimeric dihydoxyflavan-(epi)catechin proanthocyanidins have only rarely been described, mainly in plants of the genus *Senna* [7,8,9,10,11]. Recently, our research group characterized proanthocyanidins, built with flavan and flavan-3-ol monomeric units, from *D. monetaria* using high-resolution mass spectrometry [12].

Nuclear magnetic resonance (NMR) is widely used for the identification of proanthocyanidins. However, it is difficult to determine the positions of interflavan bonds, such as 4→8 or 4→6 [13]. Recently, new studies demonstrated a successful approach to identify and determine both the relative and the absolute configuration of such compounds on the basis of NMR chemical shifts and experimental electronic circular dichroism (ECD) spectra [14,15,16]. In the work presented here, we compared the calculated and experimental ECD spectra in addition to extensive NMR studies in order to determine the complete structure of a proanthocyanidin.

## 2. Results and Discussion

### 2.1. Compound Identification

Compound **1** was isolated as a reddish-brown amorphous powder. The molecular formula C_30_H_26_O_9_ was determined by HR-ESI-MS as *m*/*z* 529.1504 ([M − H]^−^ calculated for C_30_H_25_O_9_^−^ 529.1504). HR-ESI-CID-MS^2^ resulted in fragments *m*/*z* 511.1402, 419.1140, 409.0932, and 289.0721 (Appendix A) indicating compound **1** to be a dimeric proanthocyanidin (Appendix A). The constitution and relative configuration of 1 were determined by detailed 1D- and 2D-NMR measurements (Appendix A). The ^13^C-NMR spectrum (Appendix A) revealed two signal sets with 30 signals each. Despite partial signal overlap in the ^1^H-NMR spectrum (Appendix A), the proton signals could also be assigned to the two signal sets using the HSQC spectrum (Appendix A). Since all ^1^H-NMR signals of the two sets had the same correlations in the COSY, TOCSY, ROESY, and HMBC 2D-NMR spectra (Appendix A), and the ^1^H−^1^H coupling constants were almost the same for both sets, two atropisomers were present. Rotational isomerism in proanthocyanidins often results in broad NMR signals [13,14,16]; however, for compound **1**, both atropisomers were apparently stable at room temperature in the solvent used (CD_3_OD). From the quantitative evaluation of the ^1^H-NMR spectrum, a molar ratio of 1.6:1 was obtained for the major and minor atropisomers. From the COSY and TOCSY spectra, as well as the analysis of ^1^H signal multiplicities and coupling constants, four aromatic spin systems could be assigned (for the numbering scheme, see Figure 1): an AA’BB’ system (H-B2′/6′, H-B3′/5′), two tri-substituted aromatic rings (H-A5, H-A6, H-A8 and H-E2′, H-E5′, H-E6′), and one isolated proton (H-D6). Furthermore, two heterocyclic four-spin systems could be identified: H-C2, H-C3A, H-C3B, H-4 andH-F2, H-F3, H-F4A, H-F4B, with oxygen substitution at C-C2, C-F2, and C-F3, as shown by the corresponding ^1^H and ^13^C chemical shifts (Table 1 and Appendix A). The assignment of the six-spin systems to rings A, B, C, D, E, and F was possible on the basis of the HMBC and NOE correlations (see Table 1; Figure 1). H-C2 showed HMBC correlations to C-B1′ and C-B2′/6′; both protons attached to C3 to C-A10, while H-C4 showed correlations to C-A5, C-A9, C-A10. On the other hand, HMBC correlations of H-F2 to C-E1′ and C-E2′/6′, of H-F3 to C-F10, and of both protons at F4 to C-D5, C-D9, and C-D10 were identified. The linkage of the two molecular parts followed from the HMBC correlation of H-C4 to C-D7, C-D8, C-D9 on the one hand and the correlation of H-F2 to C-D9 on the other hand. Due to the large vicinal coupling constants, both protons, H-C2 and H-C4, were axially oriented (^3^J_H-C2/H-C3A_ = 11.6 Hz; ^3^J_H-C4/H-C3A_ = 12.0 Hz). H-F2 was also axially aligned, resulting from the NOE to one of the protons at F4 (H-F4 at 2.846 ppm). Since the coupling constant between the axial proton H-F4A and H-F3 was small (^3^J_H-F3/H-F4A_ = 4.6 Hz), the latter was equatorially aligned.

As result, the analysis of the 2D-NMR spectra allowed the assignment of all ^1^H and ^13^C chemical shifts for both atropisomers (see Appendix A) and the determination of the relative configuration at the four stereo centers, but it could not be determined whether the major isomer had a P or M configured biaryl axis.

### 2.2. Determination of the Absolute Configuration and the Biaryl Position by Means of ECD Calculation

The NMR results showed the protons at C2, C4, and F2, as well as the OH group at F3, to be axially oriented (see Figure 1); thus, the relative configuration at the four stereogenic centers was determined. However, according to NMR measurements on the nonderivatized compound, neither the absolute configurations nor the atropisomers could be deduced, due to missing NOE interactions between the upper and lower parts of the proanthocyanidin. Therefore, combined calculations of electronic circular dichroism (ECD) spectra and experimental values were used to determine both the absolute configuration and the atropisomers. Additionally, the proposed position of the biaryl linkage (C4–D8 instead of C4–D6; see Figure 2) could be proven.

Since the rotation about the C4–D8 bond is hindered by an energy barrier of 22.9 kcal/mol, the ECD spectra for each of the theoretically possible four stereoisomers C2*R*/C4*R*/F2*R*/F3*R*, C2*R*/C4*R*/F2*S*/F3*S*, C2*S*/C4*S*/F2*R*/F3*R*, and C2*S*/C4*S*/F2*S*/F3*S* (for numbering scheme see Figure 2), referred to in this paper as *RRRR*, *RRSS*, *SSRR*, and *SSSS*, respectively, were calculated with two alternative atropisomers each. Thus, altogether, 16 calculated ECD spectra (eight each for C4–D8 and C4–D6 connections) were compared with the experimental spectrum (see Table 2 for an overview).

Below, we discuss in detail only the ECD spectra for the nine isomers with the lowest conformational energy (Figure 1 and Figure 2 and Appendix A). The ECD spectra and structures of the related energetically disfavored atropisomers are shown in the Appendix A.

In general, the analyses carried out showed that the calculated and experimental ECD spectra of isomers with a C4–D8 connection showed subtle differences when compared with those with a C4–D6 connection. This could be a result of many different factors, such as solution state effects and level of theory, as reported by Stephens et al. [17]. Nevertheless, the results presented below show that the method used in this work was successful in assigning the structure (mostly likely) of compound **1**.

The calculated ECD spectrum that fit best with the experimental one resulted in a structure with C4–D8 connectivity with the P atropisomer *RRRR*-configuration (Figure 1A). For comparison, the figure includes the mirrored spectrum for the enantiomer (blue curve for *SSSS*-M) to visualize the difference between both. The superposition (with low similarity 0.6071) of the *SSSS*-M spectrum with the experimental one is displayed in the Appendix A. The very high similarity of 0.9674 with a low shift of 19 nm between the calculated and experimental ECD spectra is a strong indication that this configuration represents the true experimental structure, particularly when compared with the much lower similarity of the other calculated spectra for C4–D8 connectivity (Appendix A). However, for the M atropisomer, the calculated and experimental ECD spectra also fit quite nicely (similarity 0.9641) with the C4–D8 connection and *RRRR* configuration (Figure 1C), but with a significantly higher shift of 30 nm. The NMR spectra indicated the occurrence of both atropisomers. The M atropisomer had a relative energy of 1.7 kcal/mol, resulting in a Boltzmann weighted distribution of P/M = 19:1. ^1^H-qNMR data showed a relative ratio of 1.6:1 for the two atropisomers. From our ECD calculations, taking into consideration the slightly higher conformational energy and the higher shift of the calculated spectrum when superimposed with the experimental one for the M atropisomer, it seems that the P atropisomer was the preferred one, even though the molar ratio differed from that determined by qNMR. The reason could be missing entropy contributions for the free-energy calculations or the choice of insufficiently high basis sets for the DFT energy calculations. The Boltzmann weighted sum of both spectra resulted in a slightly enhanced similarity to the experimental one of 0.9682, but the fitted curves were almost identical to those shown in Figure 1A. The comparison of the calculated ECD spectra with the experimental one for the structures with C4–D6 connectivity (Figure 2 and Appendix A) showed a high similarity of 0.9278 only for the energetically disfavored (2.3 kcal/mol) P atropisomer with *RRRR* configuration (Figure 2), but a shift of 27 nm. The unfavorable energy and the high shift make it very unlikely that this structure represents the true one. All other ECD spectra for structures with C4–D6 connectivity did not fit with the experimental one.

Thus, in conclusion, on the basis of the comparison of all 16 calculated ECD spectra with the experimental ones, including the consideration of relative conformational energies, the most likely structures shown in Figure 1 were characterized by C4–D8 connectivity and *RRRR* configuration. Accordingly, compound **1** is entcassiaflavan-(4β→8)-epicatechin. This compound was described by Coetzee et al. [7], and it was isolated from *Senna petersiana* as a 4′,7-di-*O*-methyl-ent-cassiflavan-(4β→8)-3′,4′,5,7-tetra-*O*-methyl-3-*O*-acetylepicatechin derivative. 

## 3. Materials and Methods

### 3.1. General Experimental Procedures

The 1D (^1^H, ^13^C) and 2D (^1^H, ^13^C gHSQCAD, ^1^H, ^13^C gHMBCAD, ^1^H, ^13^C gH2BCAD, ^1^H, ^1^H DQFCOSY, ^1^H, ^1^H zTOCSY, ^1^H, ^1^H ROESYAD) NMR spectra were measured with an Agilent VNMRS 600 instrument at 599.83 MHz (^1^H) and 149.84 MHz (^13^C) using standard CHEMPACK 8.1 pulse sequences implemented in the VNMRJ 4.2A spectrometer software. All spectra were obtained with CD_3_OD as solvent at +25 °C. ^1^H and ^13^C chemical shifts were referenced to internal hexamethyl disiloxane (δ_H_ 0.062 ppm; δ_C_ 1.96 ppm), with the following parameters: TOCSY mixing time = 80 ms; ROESY mixing time = 300 ms; HSQC optimized for 1JCH = 146 Hz; HMBC optimized for nJCH = 8 Hz.

High-resolution ESI-MS^n^ (*m*/*z* 100 to 2000) analyses were performed by direct injection into an Orbitrap Elite Mass spectrometer (Thermofisher Scientific, Waltham, MA USA), and HR-ESI-CID-MS^2^ analyses were performed using a collision energy dissociation (CID) of 25 eV, in a negative mode of ionization. Data were acquired and processed using the Xcalibur^®^ 2.2 software.

ECD data were acquired on a JASCO J-815 CD spectrometer (solvent: methanol).

### 3.2. Extraction and Isolation of Compound

The subfraction SL6–5 (27 mg) was obtained from the ethyl acetate fraction of *Dalbergia monetaria* L. leaves from our previous study [12]. The subfraction was purified using preparative high-performance liquid chromatography (HPLC, Knauer system coupled with a WellChrom K-1001 pump and WellChrom K-2501 UV detector) using an ODS-A column (5 μm, 120 Å, 150 × 10 mm ID, YMC, USA). The mobile phase was H_2_O (A) and MeOH (B; Fluka Analytical, HPLC-MS grade Chromasolv^®^) (acidified 0.1% formic acid, *v*/*v*). The flux rate was 16 mL/min. The gradient was as follows: 2–30% B, 0 to 15 min; 30–70% B, 15 to 25 min. An isocratic condition was established (70% B) for 2 min before the reverse gradient of 2% B for 3 min. Compound **1** (14 mg) eluted at 13.49 min.

### 3.3. Computational Methods

All structures were constructed using the Molecular Operating Environment (MOE) software [18]. LowMode molecular dynamics simulations were applied for conformational search using the MMFF94 molecular mechanics force field [19]. Except for two alternative conformations of each phenolic hydroxyl group, only two low-energy conformations (atropisomers) resulted. The force field minimum-energy structures were subsequently optimized by applying density functional theory (DFT) using the BP86 functional with the def2-TZVPP basis set [20,21,22,23,24] implemented in the ab initio ORCA 3.0.3 program package [25]. The influence of methanol solvent was included in the DFT calculations using the COSMO model [26]. For the estimation of the rotational barrier for C4–D8 connectivity (Figure 1), a conformational scan around the C4–D8 bond in steps of 5° was performed. The quantum chemical simulations of the UV and ECD spectra were also carried out using ORCA. For this purpose, the first 50 excited states of each enantiomer and conformation were calculated by applying the long-range corrected hybrid functional TD CAM-B3LYP with the def2-TZVP(-f) and def2-TZVP/J basis sets [22,23,24]. The ECD curves were visualized with the help of the software SpecDis 1.64 [27,28] from the calculated rotatory strength values using a Gaussian distribution function at a half-bandwidth of σ = 0.3 eV. In all cases, the ECD spectra were superimposed with the experimental one to reach maximal similarity using SpecDis. A maximum shift of ±30 nm was allowed. Alternative superpositions based on UV spectra gave no reasonable results because of the rather flat curve of the experimental one (Appendix A). Since enantiomers show mirrored spectra, the calculated spectra of the opposite configurations listed in Table 2 were mirrored and additionally superposed with the experimental ones. Since alternative hydroxyl group conformations influenced the calculated ECD spectra only minimally, and the energies between atropisomers were mostly more than 2 kcal/mol different from each other, the energetically unfavored conformations contributed less than 5% to Boltzmann statistical weights. Therefore, only single spectra were compared with the experimental spectrum except when discussing the relevant structures related to Figure 1.

## 4. Conclusions

For the first time, entcassiflavan-(4β→8)-epicatechin (1) was isolated from a plant of the genus *Dalbergia*, from *Dalbergia monetaria* L.f. This is also the first isolation of the underivatized compound. The constitution and relative configuration were determined by detailed MS and NMR studies. All ^1^H and ^13^C signals for the two atropisomers could be assigned.

By comparison of the experimental electronic circular dichroism (ECD) spectra with the calculated ECD spectra, for the 16 reminiscent possible stereoisomers (post NMR), the absolute configurations concerning the four stereocenters could be determined. In addition, this method confirmed the position of the interflavan bond and assigned the main compound (most likely) to be the P atropisomer. Thus, we could demonstrate that, without considerable effort, e.g., by derivatization, breakdown, or synthesis of a parent compound, the absolute configuration, including atropisomerism, could be determined on the basis of NMR data and ECD spectral calculations in comparison with the experimental spectrum.

## Data Availability

The data presented in this study are openly available free of charge from the RADAR repository (https://www.radar-service.eu/radar/en/home) and can be accessed at https://dx.doi.org/10.22000/528. The data can also be requested from the corresponding authors.

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
