# Peer review of "Structural Elucidation of an Atropisomeric Entcassiflavan-(4β→8)-Epicatechin Isolated from Dalbergia monetaria L.f. Based on NMR and ECD Calculations in Comparison to Experimental Data"

_molecules, 2022, doi:10.3390/molecules27082512_

Round 1

Reviewer 1 Report

This work by Wessjohann and coworkers describes the isolation of the dihydroxyflavan-epicatechin  proanthocyanidin, entcassiflavan-(4β→ 8)-22 epicatechin from the plant Dalbergia monetaria and its structural characterization. The relative configuration was established by NMR analysis while attempt to establish its absolute configuration was carried out by computation of ECD spectrum.

However, the results of ECD computations appear questionable. In fact, the computed ECD spectra for the enantiomeric pairs RRRRP/SSSSM, RRRRM/SSSSP, SSRRP/RRSSM, and SSRRM/RRSSP for both regioisomers C4-D8 and C4-D6 should be in a mirror image relationship while, on the contrary, spectra reported in the paper are not (see for example Figure 1A and 2). Moreover, even if the computed ECD spectrum for the RRRRP stereoisomer displays a better similarity factor with experimental, the computed ECD spectrum for its atropisomer RRRRM visually fits better the experimental, reproducing the shoulder at 240 nm. The similarity factor can be useful to numerically reveal the spectral match, but it should be used with judgment. Also, the spectral shift is not a good parameter to judge the spectral match because TDDF computations often overestimate or underestimate transition energies depending on the functional and basis set employed.

The ECD analysis is the core of the paper, therefore I do not believe this paper suitable for publication in Molecules.

Minor points.

-The authors should specify the similarity protocol employed and report proper citations.

-Figures 1-8 should be condensed or some of them moved to Supplementary information. For example, panels with up to four spectra could be employed and computed structures moved to SI.

-At line 435 the authors say that “Only single spectra were compared with the experimental”. While it is reasonable the hydroxyl groups conformations do not influence the ECD spectra, conformations ascribable to aryl rings rotations should. Therefore, their contribution should be considered in the Boltzmann average.

-The ECD spectrum of compound 1 appears quite different to those of structurally similar dimeric proanthocyanidins reported in ref. 7. The authors should check this discrepancy. Do they compared NMR data reported for compounds in ref. 7 with that of 1?

Author Response

Response to Decision Letter: Molecules

Manuscript ID: molecules-1602333

Title: “Structural elucidation of an atropisomeric entcassiflavan-(4β®8)-epicatechin isolated from Dalbergia monetaria L.f based on NMR, and ECD calculations in comparison to exper-imental data”

REVIEWER 1

This work by Wessjohann and coworkers describes the isolation of the dihydroxyflavan-epicatechin  proanthocyanidin, entcassiflavan-(4β→8)-epicatechin from the plant Dalbergia monetaria and its structural characterization. The relative configuration was established by NMR analysis while attempt to establish its absolute configuration was carried out by computation of ECD spectrum.

However, the results of ECD computations appear questionable. In fact, the computed ECD spectra for the enantiomeric pairs RRRRP/SSSSM, RRRRM/SSSSP, SSRRP/RRSSM, and SSRRM/RRSSP for both regioisomers C4-D8 and C4-D6 should be in a mirror image relationship while, on the contrary, spectra reported in the paper are not (see for example Figure 1A and 2).

We are extremely grateful for this reviewer to point this out. There is not a mistake in the calculations, but through the versions of the manuscript preparation, the numbers of figures got mixed up, and thus Fig. 2 is identical to fig 1C. By this shift of figures – not accompanied by a shift of the legends, a serious error occurred and a wrong assignment, which has been corrected now.

Moreover, even if the computed ECD spectrum for the RRRRP stereoisomer displays a better similarity factor with experimental, the computed ECD spectrum for its atropisomer RRRRM visually fits better the experimental, reproducing the shoulder at 240 nm.The similarity factor can be useful to numerically reveal the spectral match, but it should be used with judgment. Also, the spectral shift is not a good parameter to judge the spectral match because TDDF computations often overestimate or underestimate transition energies depending on the functional and basis set employed.

The reviewer is completely right with all. We slightly modified the text to, but we have to hint that in the final conclusion we only stated that the experimental structure is clearly RRRR with C4-C8 connectivity, without focusing on the predicted atropisomer where we see P to be more likely than M.

The ECD analysis is the core of the paper, therefore I do not believe this paper suitable for publication in Molecules.

Minor points.

-The authors should specify the similarity protocol employed and report proper citations.

We added corresponding information in the methods part (3.3 Computational Methods).

-Figures 1-8 should be condensed or some of them moved to Supplementary information. For example, panels with up to four spectra could be employed and computed structures moved to SI.

Thank you very much for that suggestion, we have rearranged the figures. Now, in the main manuscript, only three images are presented (Figures 1-3), referring to the ECD spectra of analyzed isomers. The structures of each one were moved to the supporting information (Figures S9-S11).

-At line 435 the authors say that “Only single spectra were compared with the experimental”. While it is reasonable the hydroxyl groups conformations do not influence the ECD spectra, conformations ascribable to aryl rings rotations should. Therefore, their contribution should be considered in the Boltzmann average.

We compared the spectra with alternative directions of the hydroxyl groups, but with very minor alterations: they are identical to each other thus a Boltzmann average makes no sense because no other results will appear in all: shape of the curve, similarity, and shift.

-The ECD spectrum of compound 1 appears quite different to those of structurally similar dimeric proanthocyanidins reported in ref. 7. The authors should check this discrepancy. Do they compared NMR data reported for compounds in ref. 7 with that of 1? 

Coetzee et al. (ref. 7 of the manuscript) described the occurrence of several dimeric proanthocyanidins, including the one described in our manuscript. However, Coetzee et al. did not isolate the underivatized compounds, but rather the permethyl aryl ether acetate derivatives ("The mixture could be resolved by gel column chromatography and the compounds purifed by TLC as their permethylaryl ether acetate derivatives 2, 4, 6, 8, 10, 12 and 14, respectively."). The NMR data in ref. 7 are therefore given only for the derivatized compounds, and deviations in the chemical shifts are therefore to be expected. Also, different solvents were used in the NMR measurements (CDCl3 vs. CD3OD).

Thank you so much,

Sincerely,

Profa. Ivana Correa Ramos Leal

Laboratório de Produtos Naturais e Ensaios Biológicos, Faculdade de Farmácia – Universidade Federal do Rio de Janeiro. Avenida Carlos Chagas Filho - CCS, Cidade Universitária, Rio de Janeiro – RJ. CEP: 21941-902. +55 21 965620428 (mobile phone)

E-mail: [email protected] (I. C. R. Leal); [email protected]

Reviewer 2 Report

A competently performed and reported study. Specialist interest only, but a sound piece of work

Author Response

Response to Decision Letter: Molecules

Manuscript ID: molecules-1602333

Title: “Structural elucidation of an atropisomeric entcassiflavan-(4β®8)-epicatechin isolated from Dalbergia monetaria L.f based on NMR, and ECD calculations in comparison to exper-imental data”

REVIEWER 2

A competently performed and reported study. Specialist interest only, but a sound piece of work.

We thank you for your review and compliments. Regarding to the manuscript, we have added some more information and, with the aim of improving it, we proceeded some additional adjustments.

Thank you so much,

Sincerely,

Profa. Ivana Correa Ramos Leal

Laboratório de Produtos Naturais e Ensaios Biológicos, Faculdade de Farmácia – Universidade

Federal do Rio de Janeiro. Avenida Carlos Chagas Filho - CCS, Cidade Universitária, Rio de

Janeiro – RJ. CEP: 21941-902. +55 21 965620428 (mobile phone)

E-mail: [email protected] (I. C. R. Leal); [email protected]

Reviewer 3 Report

Peer-Review of Molecules Manuscript 1602333

The manuscript entitled “Structural elucidation of an atropoisomeric entcassiflavan-(4-8)-epicatechin isolated from Dalbergia monetaria L.f based on ECD calculations in comparison to experimental data” describes the use of NMR and ECD spectroscopy to ascertain the 3-D structure of the title compound.

The work described is mostly well presented, accurately supported by the included references and supporting information, and conclusions are in line with results obtained.

Nevertheless, important issues in the submitted manuscript need to be addressed before publication in Molecules is granted.

  • Firstly, the title given to the manuscript do not represent correctly the work described. While ECD is used to obtain some structural features, NMR is the main technique used to elucidate the chemical structure of 1, including the relative configuration. The title needs to be re-written to accurately describe this.

  • As clearly stated in the manuscript (first paragraph, page 4), the 2D structure and relative configuration of 1 can be deduced from the detailed NMR analysis done. Nevertheless, the authors decided to use ECD to further test these findings, along with stablishing the absolute configuration. In my opinion, the methodology chosen by the authors can safely ascertain the AC of 1, but ECD similarities do not provide solid evidence to differentiate between several stereoisomers. It’s clear from figures 1-8 that the ECD spectra of compounds with the C4-D8 bond are very similar, and only more important differences appear when compared with those with the C4-D6 bond. The more subtle differences, between the observed and calculated ECD spectra of isomers with the same connectivity, can be produced by many different factors, such as level of theory, solution state effects, etc., and can not be trusted to propose a full stereochemical determination. Even in cases when a relative configuration is secured, these small differences are common, and the use of more than one chiroptical method is advised. For a more detailed explanation of ECD limitations, I recommend the article of Stephens et.al.: https://doi.org/10.1021/jo062567p.

  • In the same lines, its not clear to me how wavelength shifts were obtained for the comparison showed. I assume that shifts were calculated by the SpecDis software to obtain the optimal similarity between calculated and observed spectra. Which spectrum, UV or ECD, was used to obtain the shifts?

  • The authors report a discrepancy between the abundances of both atropisomers observed with qNMR and those calculated using DFT free energies. I recommend using more recent levels of theory to calculate thermochemical parameters that might produce better results, such as the recently developed r2-SCAN-3C.

Author Response

Response to Decision Letter: Molecules

Manuscript ID: molecules-1602333

Title: “Structural elucidation of an atropisomeric entcassiflavan-(4β®8)-epicatechin isolated from Dalbergia monetaria L.f based on NMR, and ECD calculations in comparison to exper-imental data”

REVIEWER 3

The manuscript entitled “Structural elucidation of an atropoisomeric entcassiflavan-(4-8)-epicatechin isolated from Dalbergia monetaria L.f based on ECD calculations in comparison to experimental data” describes the use of NMR and ECD spectroscopy to ascertain the 3-D structure of the title compound.

The work described is mostly well presented, accurately supported by the included references and supporting information, and conclusions are in line with results obtained.

Nevertheless, important issues in the submitted manuscript need to be addressed before publication in Molecules is granted.

Firstly, the title given to the manuscript do not represent correctly the work described. While ECD is used to obtain some structural features, NMR is the main technique used to elucidate the chemical structure of 1, including the relative configuration. The title needs to be re-written to accurately describe this.

Thank you for this observation. The title has been slightly modified and now is: “Structural elucidation of an atropisomeric entcassiflavan-(4β→8)-epicatechin isolated from Dalbergia monetaria L.f based on NMR, and ECD calculations in comparison to experimental data.

As clearly stated in the manuscript (first paragraph, page 4), the 2D structure and relative configuration of 1 can be deduced from the detailed NMR analysis done. Nevertheless, the authors decided to use ECD to further test these findings, along with stablishing the absolute configuration. In my opinion, the methodology chosen by the authors can safely ascertain the AC of 1, but ECD similarities do not provide solid evidence to differentiate between several stereoisomers. It’s clear from figures 1-8 that the ECD spectra of compounds with the C4-D8 bond are very similar, and only more important differences appear when compared with those with the C4-D6 bond. The more subtle differences, between the observed and calculated ECD spectra of isomers with the same connectivity, can be produced by many different factors, such as level of theory, solution state effects, etc., and can not be trusted to propose a full stereochemical determination. Even in cases when a relative configuration is secured, these small differences are common, and the use of more than one chiroptical method is advised. For a more detailed explanation of ECD limitations, maybe it should be cited but not more is possible? I recommend the article of Stephens et.al.: https://doi.org/10.1021/jo062567p.

In principle the reviewer is right to calculations of ECD spectra have to be considered with caution. However, the differences particularly the similarities (and shifts) are clearly lower for the other configurations even when comparing only 4-8 connectivity (see table 2, for RRRR higher than 0.96 and for all other lower than 0.86. We feel this is significant and out of the approaches and uncertainty of the methods used. Concerning your consideration, we cited the proposed reference in the main text and we have inserted a new paragraph: “In general, the analyzes carried out show that the calculated and experimental ECD spectra of isomers with C4-D8 connection showed subtle differences among them, while compared with those with C4-D6 connection. It can be result by many different factors, such as solution state effects, level of theory, as reported by Stephens et al. Nevertheless, the results presented below show that the method used in this work was successful in as-sign the structure (mostly likely) of compound 1.” (page 6)

In the same lines, its not clear to me how wavelength shifts were obtained for the comparison showed. I assume that shifts were calculated by the SpecDis software to obtain the optimal similarity between calculated and observed spectra. Which spectrum, UV or ECD, was used to obtain the shifts?

Indeed this information was missing. We added corresponding information in the methods part, as you can see at 3.3. Computational methods: “In all cases the ECD spectra were superposed with the experimental one to reach maximal similarity using SpecDis. A maximum of ± 30 nm shift was allowed. Since enantiomers show the mirrored spectrum the calculated spectra of the configurations listed in Table 2 left were mirrored and additionally superposed with the experimental one.”

The authors report a discrepancy between the abundances of both atropisomers observed with qNMR and those calculated using DFT free energies. I recommend using more recent levels of theory to calculate thermochemical parameters that might produce better results, such as the recently developed r2-SCAN-3C.

During the performance of the calculations (more than one year ago) this function was not yet available. Now two of the authors are already in pension and we are thus unable to repeat all these calculations. We very much hope the reviewer will accept this as (hopefully) there always will be a better method in yet another year etc. In particular it is only a very minor aspect of the paper to reproduce the correct ratio between P and M atropisomer of RRRR, as depending on temperature and time these are interconverting. Here we were just lucky that this process is slow on the NMR timescale.

Thank you so much,

Sincerely,

Profa. Ivana Correa Ramos Leal

Laboratório de Produtos Naturais e Ensaios Biológicos, Faculdade de Farmácia – Universidade Federal do Rio de Janeiro. Avenida Carlos Chagas Filho - CCS, Cidade Universitária, Rio de Janeiro – RJ. CEP: 21941-902. +55 21 965620428 (mobile phone)

E-mail: [email protected] (I. C. R. Leal); [email protected]

Reviewer 4 Report

The manuscript describes isolation and structure determination of poliphenolic compound from Amazon origin plant, named entcassiflavan-(4beta-8)-epicatechin. This compound possess four stereogenic centers and one chirality axis. The structure has been established by means of MS and NMR. This part of manuscript is convincing and I  have only one comment.  It is not clear why HMBC correlations C4-H to D8 and D9 are not  enough to exclude right structure (C4-D6) on Scheme 2. However the ECD part arises more doubts. 
   First, the authors compare experimental ECD spectrum of a mixture of two atropisomers (1:1.6), with calculated ECD spectrum of single atropisomer. Even though the calculated spectra of both rotamers are similar, this is not the right way. Atropisomers should be separated (HPLC) or calculated ECD spectra of M and P isomers should be mixed in the same ratio (1:1.6), and then compared. 
I do not understand why enantiomers C4-D8 like RRRR-M and SSSS-P (Table 2) have different energy (relatively 1.7 and 2.9 kcal/mol) ?  That suggests  errors in optimalisation. 
Why enantiomers (mirror images) like RRRR-P and SSSS-M (Fig. 1A,B and 2) have the same sign in ECD.  
The energy comparison of  structures with different connectivities (C4-D8 and C4-D6) in Table 2 is questionable. Also the "shift" in Table 2 needs explanation. Is it UV shift ?
I am not sure if ECD calculation is a good method for such complicated, polar compounds like flavans and catechins. I would also suggest (if possible) application of  VCD that gives narrow bands  (please check Polavarapu, P. L. Why is it important to simultaneously use more than one chiroptical spectroscopic method for determining the structures of chiral molecules? Chirality 2008, 20, 664).
 I think that  final conclusion is to optimistic.  
Beside the above comments, here are some minor points.
1. Scheme 1. It would be nice to have structure with stereogenic centers like scheme in SI. 
2. Scheme 1. Proton F3 is missing.
3. Scheme 2. The structures should possess  numbers/letters instead of right-left structure. 
4. Line 151. I am not sure if the value of energy barrier (22.9 kcal/mol) has been calculated ?
5. Line 153.  D2R/D3R etc should be corrected to F2R/F3R etc. Ring D is aromatic. 
6. Why the structures optimization was done by use of the BP86 functional while the ECD spectra were calculated using the CAM-B3LYP one?
7. How the similarity factor was calculated? 
8. Remove from SI  "Error! Bookmark not defined. "
9. I guess 13C NMR was recorded at 150 MHz, not 125 MHz.
10. Spelling should be carefully corrected. For example in title: "atropoisomeric" , line 189, 194 and 239 "Bolzmann". References 1-9 have double  numbers.

Author Response

Response to Decision Letter: Molecules

Manuscript ID: molecules-1602333

Title: “Structural elucidation of an atropisomeric entcassiflavan-(4β®8)-epicatechin isolated from Dalbergia monetaria L.f based on NMR, and ECD calculations in comparison to exper-imental data”

REVIEWER 4

The manuscript describes isolation and structure determination of poliphenolic compound from Amazon origin plant, named entcassiflavan-(4beta-8)-epicatechin. This compound possess four stereogenic centers and one chirality axis. The structure has been established by means of MS and NMR. This part of manuscript is convincing and I  have only one comment.  It is not clear why HMBC correlations C4-H to D8 and D9 are not  enough to exclude right structure (C4-D6) on Scheme 2. However the ECD part arises more doubts. 
   First, the authors compare experimental ECD spectrum of a mixture of two atropisomers (1:1.6), with calculated ECD spectrum of single atropisomer. Even though the calculated spectra of both rotamers are similar, this is not the right way. Atropisomers should be separated (HPLC) or calculated ECD spectra of M and P isomers should be mixed in the same ratio (1:1.6), and then compared.

In principle the reviewer is right and indeed we performed a Boltzmann distribution but because both spectra are almost identical (see text and figures) we did refrain for adding an additional figure which delivers no further information. Also we agree that the atropisomers are differentiable on the NMR timescale, but unfortunately this does not necessarily mean they are separable by chromatography with its much longer time frame. At least, we – like others - were unable to separate the atropisomers.

I do not understand why enantiomers C4-D8 like RRRR-M and SSSS-P (Table 2) have different energy (relatively 1.7 and 2.9 kcal/mol) ?  That suggests  errors in optimalisation. 
Why enantiomers (mirror images) like RRRR-P and SSSS-M (Fig. 1A,B and 2) have the same sign in ECD.

This was indeed an error. Especially because we created the spectra from the mirror of the enantiomer. This was now corrected and a clearer Table 2 has been created (on page 6).

The energy comparison of  structures with different connectivities (C4-D8 and C4-D6) in Table 2 is questionable.

This seems to be a misunderstanding. The given energies are relative energies between the atropisomers with identical connectivities and stereochemistry. We hope with the modified Table 2 it is better understandable.

Also the "shift" in Table 2 needs explanation. Is it UV shift ?

It is related to ECD shift. Corresponding explanation was added it in the methods section.

I am not sure if ECD calculation is a good method for such complicated, polar compounds like flavans and catechins. I would also suggest (if possible) application of  VCD that gives narrow bands  (please check Polavarapu, P. L. Why is it important to simultaneously use more than one chiroptical spectroscopic method for determining the structures of chiral molecules? Chirality 2008, 20, 664).

Beside the above comments, here are some minor points.
1. Scheme 1. It would be nice to have structure with stereogenic centers like scheme in SI. 

Now the Scheme 1 has the structure with stereogenic centers assigned.

  1. Scheme 1. Proton F3 is missing.

The Scheme 1 is complete now with all protons.

  1. Scheme 2. The structures should possess  numbers/letters instead of right-left structure. 

The Scheme 2 was modified. Now letters are used to distinguish each structure.

  1. Line 151. I am not sure if the value of energy barrier (22.9 kcal/mol) has been calculated?

This has already been explained in the methods section. Was done by a rotational scan in 5° steps for estimation.

  1. Line 153.  D2R/D3R etc should be corrected to F2R/F3R etc. Ring D is aromatic. 

Thank you for this observation. The configuration previously assigned to ring D, have been corrected and now, all of them, are assigned to ring F.

  1. Why the structures optimization was done by use of the BP86 functional while the ECD spectra were calculated using the CAM-B3LYP one?

This work was done about just before my retirement (W.B.). During this time, we checked several options for functionals for optimal running time and most accurate CD-spectra calculations. This option gave best results although I agree maybe the optimizations could have done with B3LYP as well. However, after my retirement, there was no chance and time to inspect and test newer and better suited functionals for the calculations.

  1. How the similarity factor was calculated? 

Now it is explained in a 3.3. topic of Material and methods. It was performed using SpecDis.

  1. Remove from SI  "Error! Bookmark not defined. "

The phrase “"Error! Bookmark not defined. " was removed from SI.

  1. I guess 13C NMR was recorded at 150 MHz, not 125 MHz.

It is correct, the 13NMR was recorded at 125 MHz. This value has been adjusted in the text.

  1. Spelling should be carefully corrected. For example in title: "atropoisomeric" , line 189, 194 and 239 "Bolzmann". References 1-9 have double  numbers.

We thank the reviewer for the detailed suggestion and have either corrected the paper accordingly or amended the respective explanation in the text and experimental part, or we added references there that is resolves the case.

Thank you so much,

Sincerely,

Profa. Ivana Correa Ramos Leal

Laboratório de Produtos Naturais e Ensaios Biológicos, Faculdade de Farmácia – Universidade Federal do Rio de Janeiro. Avenida Carlos Chagas Filho - CCS, Cidade Universitária, Rio de Janeiro – RJ. CEP: 21941-902. +55 21 965620428 (mobile phone)

E-mail: [email protected] (I. C. R. Leal); [email protected]

Round 2

Reviewer 1 Report

In their revised paper Wessjohann and coworkers addressed some of the issues raised by the reviewer, but some important questions remain.

The main issue concern the agreement of ECD spectra in manuscript and SI. In fact, the ECD spectrum of C4-D8 SSSSM-1 reported in Figure 1A is different from that reported in Figure S8 for the same stereoisomer. The same happens for compound C4-D8 SSSSP-1, where the spectra in Figure 1C and S15 are different. Moreover, the spectra of the enantiomeric pair C4-D8 SSRRP/RRSSM should be in a mirror image relationship, while spectra in Figures S9 and S10 are not. Moreover, From Figure 1 it seem that the sign of the ECD couplet-like feature centered at 210 nm is due to the absolute configuration of the compound stereocenters and not to the atropisomerism (stereoisomers RRRRM and RRRRP have the same couplet sign). However, in ECD spectra in the SI it seems that stereoisomers with opposite absolute configuration at the flavan stereocenters display that couplet-like feature with the same sign. This is not reasonable.

When answering to the first reviewer report authors said than in the original manuscript there were errors in Figures number and that they corrected them. However, the main problems remain, probably, because also in the revised version many errors occur. I don’t know the reason for that, but the authors should check these heavy errors and solve them.

I also have some concern about the reliability of the absolute configuration assignment. The authors reported that computed ECD spectra are wavelength shifted to increase the similarity factors with the experimental ECD. However, the wavelength shift to apply should be chosen to achieve the maximum coincidence of computed and experimental UV spectra maxima. Experimental and computed UV spectra are not reported in the Figures, therefore it is not possible to verify the UV agreement.

I also believe that the sum of the computed ECD spectra for the RRRRM and RRRRP stereoisomers should be carried out taking into account their 1.6:1 ratio experimentally found by qNMR. In fact, ratio obtained by computed energies is not reliable, because the authors take into account only a single conformer for each stereoisomer.

In summary, I do not believe the paper suitable for publication in Molecules in the present form and major revisions taking into account the issues raised above are needed.

Minor points.

-Line 334: …ECD spectra for eight isomers..

-Line 595 and 596: reference 24 is 25 and reference 25 is 26.

-Lines 335-336: Please, check SI Figure numbers reported in this sentence because they seem in disagreement with those reported in Table 2.

Reviewer 3 Report

Peer-Review of Molecules Manuscript 1602333

The manuscript entitled “Structural elucidation of an atropisomeric entcassiflavan-(4b→8)-epicatechin isolated from Dalbergia monetaria L.f based on NMR, and ECD calculations in comparison to experimental data” describes alternative pathways to obtain the enantiomers of the title compound.

The current re-submitted version of the article show clear improvements and addresses most of the issues pointed out from my part in a previous report. I believe that the work shown in the manuscript is of enough merit to be published in Molecules.

Reviewer 4 Report

The manuscript was improved and the conclusions were moderated. Though I am still not fully convinced (see following questions).

1) It is not clear why HMBC correlations C4-H to D8 and D9 are not  enough to exclude  structure  B (C4-D6) on Scheme II. 
2) The conclusion (page 8) shows wrong figures: 
" Thus, in conclusion, based on the comparison of the all sixteen calculated ECD spectra with the experimental ones, including consideration of relative conformational energies, the structures shown in Figures S9A and S9B are characterized by the C4-D8 connectivity and RRRR-configuration as the most likely the experimental structures."
3) Calculated ECD spectra for enantiomers D8 SSRR-M (Fig. S16) and D8 RRSS-P (Fig. S17) have the same sign. 
4) Calculated ECD spectra for enantiomers D8 RRRR-P (Fig. 1A) and D8 SSSS-M (Fig. S8) have the same sign.
5) 13C NMR spectra on 600MHz NMR spectrometer are recorded at 150 MHz (not 125 MHz). 
